Resource

# Reverse-engineering the anti-MUC1 antibody 139H2 by mass spectrometry–based de novo sequencing

Weiwei Peng[1,*], Koen CAP Giesbers[2,*], Marta Šiborová[1], J Wouter Beugelink[3], Matti F Pronker[1], Douwe Schulte[1], John Hilkens[4], Bert JC Janssen[3], Karin Strijbis[2], Joost Snijder[1]

**Mucin 1 (MUC1) is a transmembrane mucin expressed at the apical surface of epithelial cells at mucosal surfaces. MUC1 has a barrier function against bacterial invasion and is well known for its aberrant expression and glycosylation in adenocarcinomas. The MUC1 extracellular domain contains a variable number of tandem repeats (VNTR) of 20 amino acids, which are heavily O-linked glycosylated. Monoclonal antibodies against the MUC1 VNTR are powerful research tools with applications in the diagnosis and treatment of MUC1-expressing cancers. Here, we report direct mass spectrometry–based sequencing of anti-MUC1 hybridoma-derived 139H2 IgG, enabling reverse-engineering of the functional recombinant monoclonal antibody. The crystal structure of the 139H2 Fab fragment in complex with the MUC1 epitope was solved, revealing the molecular basis of 139H2 binding specificity to MUC1 and its tolerance to O-glycosylation of the VNTR. The available sequence of 139H2 will allow further development of MUC1-related diagnostic, targeting, and treatment strategies.**

## Introduction

The mucin MUC1 is a transmembrane glycoprotein expressed by epithelial cells at different mucosal surfaces including the breast tissue, airways, and gastrointestinal tract. The full-length MUC1 protein extends 200–500 nm from the apical surface of epithelial cells and is therefore an important component of the glycocalyx (1, 2). At the mucosal surface, MUC1 has an essential barrier function against bacterial and viral invasion (3, 4), but it can also be used as an entry receptor by pathogenic *Salmonella* species (5). Using knockout mice, it was demonstrated that MUC1 has anti-inflammatory functions (6, 7, 8). However, MUC1 is most well known for its aberrant expression and glycosylation in different types of adenocarcinomas (9).

The full-length MUC1 heterodimer consists of an extracellular domain with a variable number of tandem repeats (VNTR) of 20 amino acids, which are heavily O-linked glycosylated, a non-covalently attached SEA domain, a transmembrane domain, and a cytoplasmic tail with signaling capacity (see Fig 1). The VNTR region consists of repeats of 20 amino acids with the sequence GSTAPPAHGVTSAPDTRPAP (10, 11). Each repeat contains five serine and threonine residues that can be O-linked glycosylated, and experiments with synthetic MUC1 fragments demonstrated a high glycosylation occupancy at these residues (12). In healthy tissues, the O-glycans on the MUC1 VNTR predominantly consist of elongated core 2 structures, whereas it remains restricted to predominant core 1 structures in many cancerous cells (13, 14).

The overexpression and altered glycosylation of MUC1 in cancerous cells makes it a potentially viable candidate target for cancer immunotherapy. In addition, MUC1 could be an interesting target for therapeutic strategies that require delivery to the (healthy) mucosal surface. Monoclonal antibodies against the MUC1 VNTR can be powerful tools because of their multiplicity of binding and possible applications in the diagnosis and treatment of MUC1-expressing cancers. Since the late 1980s, several monoclonal antibodies against MUC1 have been described and explored for the diagnosis and treatment of MUC1-overexpressing cancers (15, 16). Peptide mapping experiments have revealed that many such monoclonal antibodies target a similar region within the VNTR of MUC1, resulting in the definition of an immunodominant peptide corresponding to the subsequence APDTRPAP (17). One such antibody is 139H2, a hybridoma monoclonal antibody that was raised against human breast cancer plasma membranes (15, 16). In different studies, 139H2 has been applied for the diagnostics of MUC1-overexpressing cancers and radioimmunotherapy (15, 16, 18). In addition, the antibody is also widely applied as a research tool in Western blot, ELISA, immunohistochemistry, and immunofluorescence microscopy to study MUC1 biology (16, 19, 20). To make this antibody available for general use, we set out to determine its sequence based on the available hybridoma-derived product.

[1]Biomolecular Mass Spectrometry and Proteomics, Bijvoet Center for Biomolecular Research and Utrecht Institute of Pharmaceutical Sciences, Utrecht University, Utrecht, Netherlands [2]Department of Biomolecular Health Sciences, Division of Infectious Diseases and Immunology, Faculty of Veterinary Medicine, Utrecht University, Utrecht, Netherlands [3]Structural Biochemistry, Bijvoet Center for Biomolecular Research, Department of Chemistry, Faculty of Science, Utrecht University, Utrecht, Netherlands [4]Division of Molecular Genetics, The Netherlands Cancer Institute, Amsterdam, Netherlands

Correspondence: k.strijbis@uu.nl; j.snijder@uu.nl
*Weiwei Peng and Koen CAP Giesbers contributed equally to this work

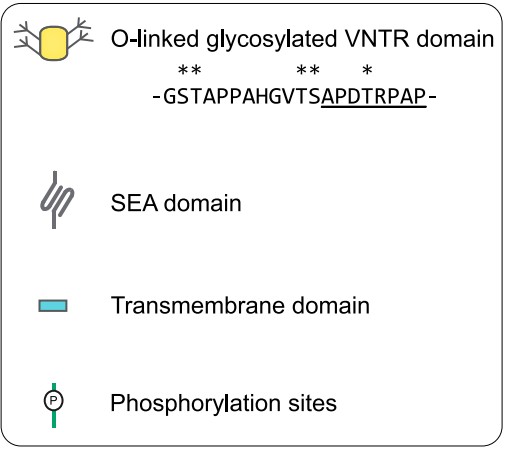

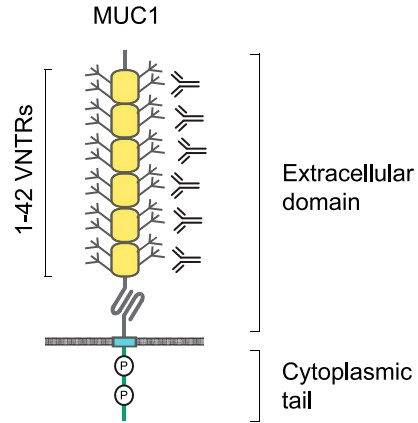

**Figure 1. Schematic overview of the MUC1 domain structure.**
VNTR, variable number of tandem repeats; SEA, domain name from initial identification in a sperm protein, enterokinase, and agrin.

Recently, we have reported a method to reverse-engineer monoclonal antibodies by determining the sequence directly from the purified protein product based on liquid chromatography coupled to mass spectrometry (LC-MS), using a bottom-up proteomics approach (21, 22, 23, 24). Here, we applied this method to obtain the full sequence of 139H2. The sequence was successfully validated by comparing the performance of the reverse-engineered 139H2 and its Fab fragment with the hybridoma-derived product in Western blot and immunofluorescence microscopy. Reverse-engineering 139H2 enabled us to characterize binding to the immunodominant peptide epitope within the MUC1 VNTR by surface plasmon resonance (SPR) and map out the epitope by solving a crystal structure of the 139H2 Fab fragment in complex with the APDTRPAP peptide. These analyses reveal the molecular basis of 139H2 binding to MUC1 and illustrate a remarkable diversity of binding modes to the immunodominant epitope in comparison with other reported structures of anti-MUC1 monoclonals targeting the VNTR.

# Result

### De novo sequencing by bottom-up mass spectrometry

The goal of our study was to obtain the sequence of the full-length 139H2 IgG antibody using a bottom-up proteomics approach. As a starting point, we used 139H2 IgG hybridoma supernatant and purified the antibody using protein G affinity resin. The purified IgG was digested with a panel of four proteases in parallel (trypsin, chymotrypsin, α-lytic protease, and thermolysin) to generate overlapping peptides for the LC-MS/MS analysis, using a hybrid fragmentation scheme with stepped high-energy collision dissociation (sHCD) and electron-transfer high-energy collision dissociation (EThcD) on all peptide precursors. The peptide sequences were predicted from the MS/MS spectra using PEAKS and assembled into the full-length heavy- and light-chain sequences using the in-house–developed software Stitch. This resulted in the identification of a mouse IgG1 antibody with an IGHV1-53 heavy chain paired with an IGKV8-30 light chain (the full sequence is provided in the Supplementary Information). The depth of coverage

for the complementarity-determining regions (CDRs) varies from around 10–100, indicating a high sequence accuracy (see Fig S1). Examples of MS/MS spectra supporting the CDRs of both heavy chain and light chain are shown in Fig 2. Comparison with the inferred germline precursors indicates a typical moderate level of somatic hypermutation (3% in the light chain; 10% in the heavy chain), with some notable mutations in the framework regions, also directly flanking CDRH2.

### Validation of the experimentally determined 139H2 sequence

The experimentally determined sequences of the 139H2 variable domains were codon-optimized for mammalian expression and subcloned into expression vectors with the mouse IgG1 heavy-chain (with an 8xHis-tag) and the kappa light-chain backbones (see Supplementary Information for the full amino acid sequences). Cotransfection of the two plasmids in HEK293 cells yielded ca. 10 mg from a 1-liter culture after HisTrap purification (see Fig S2A and B). In addition, the fragment antigen-binding (Fab) region was expressed to study the monovalent binding to MUC1. The recombinant 139H2 IgG and Fab were then compared with the hybridoma-derived 139H2 IgG in Western blot and confocal immunofluorescence microscopy.

To investigate the specificity of the recombinant 139H2 antibody for MUC1, we performed immunoblot analysis on lysates of the methotrexate-adapted human colon cancer cell line HT29-MTX, known for its high MUC1 expression, and a MUC1 knockout of the same cell line that was previously described (see Fig 3A) (5). The original hybridoma-derived 139H2 recognizes one predominant band at an estimated molecular weight of 600 kD, corresponding to full-length MUC1, and this band is absent in lysates of the MUC1-knockout cells. The recombinant 139H2 showed the same binding pattern. In confocal immunofluorescence microscopy, original hybridoma-derived 139H2 stains MUC1 at the apical surface in a confluent culture of HT29-MTX, and this signal is reduced to the background in the MUC1-knockout cell line (see Fig 3B). A similar staining is observed with the recombinant 139H2. Western blot and immunofluorescence microscopy using the monovalent Fab fragment also showed specific binding to MUC1 in the WT background

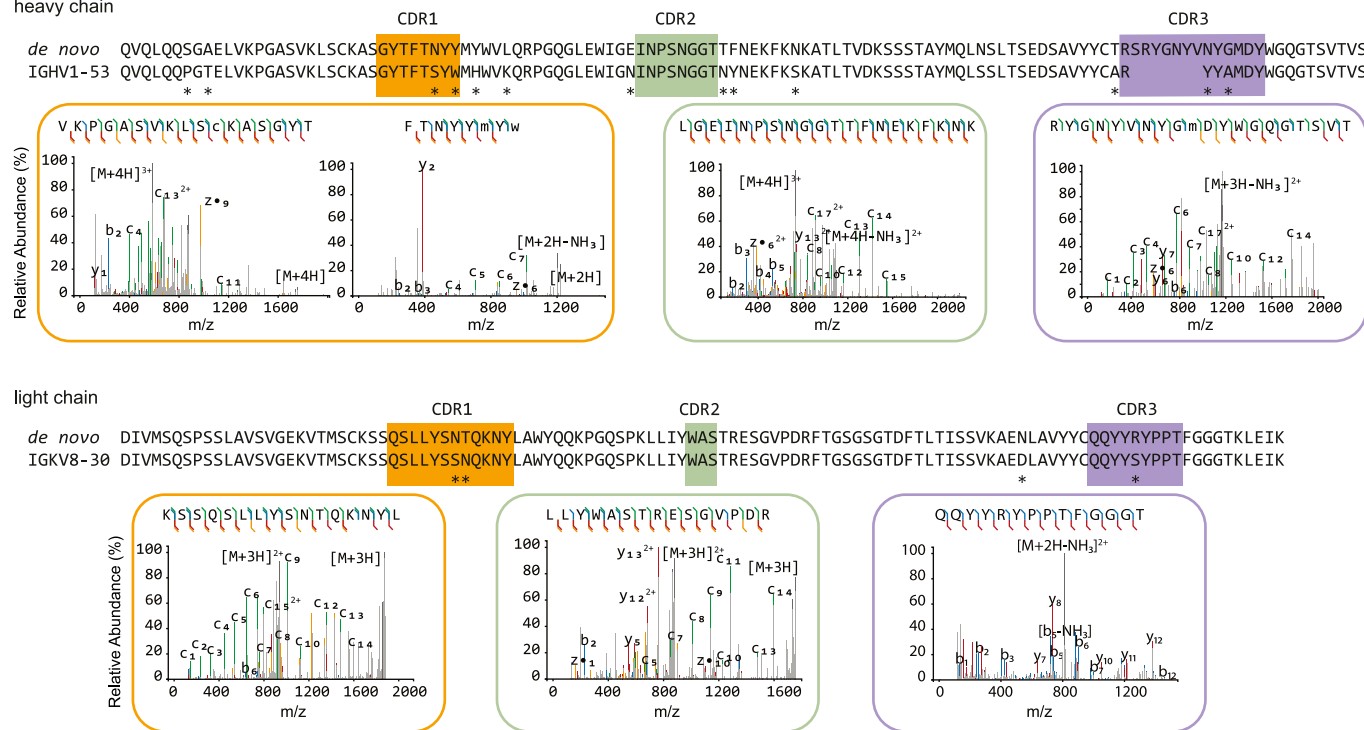

**Figure 2. De novo sequencing of the hybridoma 139H2 based on bottom-up proteomics.**
The variable region alignment to the inferred germline sequence is shown for both heavy and light chains. Positions with putative somatic hypermutation are highlighted with asterisks (*). The MS/MS spectra supporting the complementarity-determining region are shown beneath the sequence alignment, b/y ions are indicated in blue and red, whereas c/z ions are indicated in green and yellow.

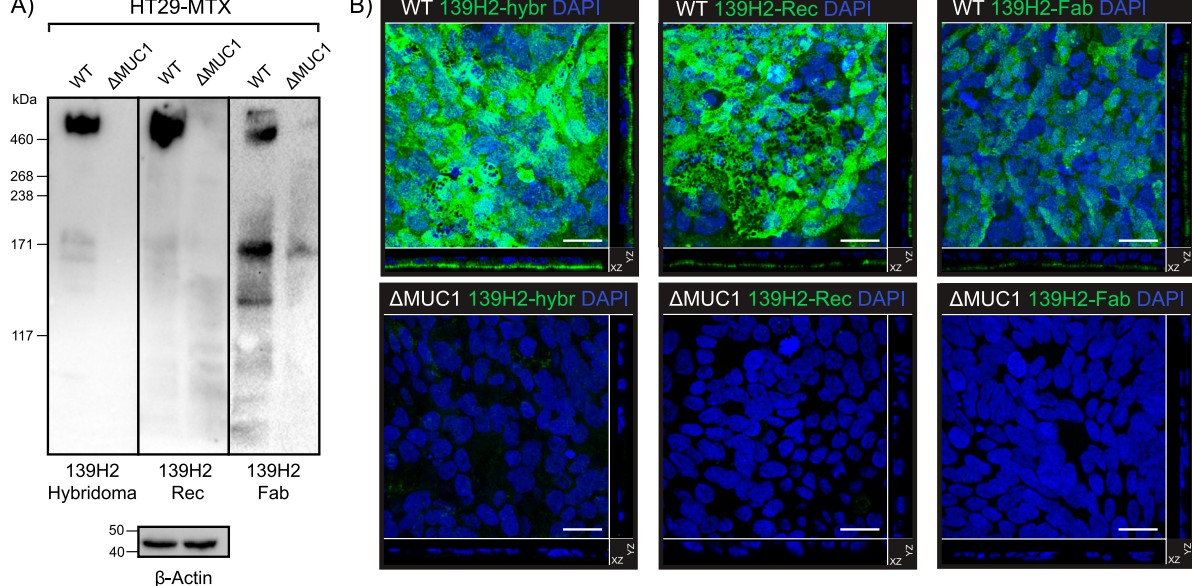

**Figure 3. Validation of synthetic recombinant 139H2 following the mass spectrometry–derived sequence.**
**(A)** Immunoblot analysis of lysates of intestinal epithelial HT29-MTX and HT29-MTX ΔMUC1 cells with the original hybridoma-derived 139H2 IgG antibody and synthetic recombinant 139H2. **(B)** Immunofluorescence confocal microscopy imaging of confluent HT29-MTX and HT29-MTX ΔMUC1 monolayers. Cells were stained for nuclei (DAPI, blue) and MUC1 (139H2, green). The signal of the 139H2 Fab was enhanced to compensate for the expected low signal/binding. White scale bars represent 20 μm.

but with reduced avidity compared with the full bivalent IgG molecule. These results confirm that the reverse-engineered 139H2 antibody is functional and recognizes the full-length MUC1 glycoprotein at the apical surface of intestinal epithelial cells.

## Epitope mapping of 139H2

Using the reverse-engineered 139H2 product, we next characterized binding to the immunodominant epitope APDTRPAPG within the MUC1 VNTR. Binding to the synthetic peptide, including an N-terminal biotin and short peptide linker for immobilization to the SPR substrate (i.e., biotin-GGS-APDTRPAPG), was determined by SPR. Binding of the full IgG was characterized by a high- and low-affinity phase with dissociation constants of $17 \times 10^{-9}$ and $43 \times 10^{-7}$ M, respectively (Fig S3A and B). We interpret this biphasic binding as an avidity-enhanced bivalent mode (both Fab arms engaged with epitope, high affinity) and a monovalent mode (single Fab arm, low affinity) of binding, respectively. In line with this interpretation, binding to a recombinant monovalent 139H2 Fab yielded a dissociation constant of $45 \times 10^{-7}$ M, similar to the low-affinity binding phase of the full IgG.

To better understand the molecular basis of 139H2 binding to the immunodominant epitope within the VNTR, we determined a crystal structure of the Fab fragment in complex with the synthetic APDTRPAPG peptide (without N-terminal biotin or peptide linker). Crystals diffracted to a resolution of 2.5 Å, and a structure was solved using molecular replacement with a ColabFold model of the 139H2 Fab. This also revealed clear density for the peptide epitope in contact with the CDRs of 139H2 (see Table S1 and Fig S4A and B).

The APDTRPAPG peptide binds diagonally across the cleft between the heavy and light chains, making direct contact with all CDRs, except CDRL2 (see Fig 4A and B and Table S2). Contact points between the peptide and the 139H2 Fab include hydrogen bonds with the peptide backbone at six of eight positions. Both the aspartic acid and arginine residues within the epitope make salt bridges with side chains from 139H2. Although D3 interacts with R99 within CDRL1, R5 interacts with E50 and T59 near CDRH2, in addition to a stacking interaction with Y100 in CDRL3. Previous studies on the binding specificity of 139H2 have shown that R5 of the epitope is crucial for 139H2 binding. The interactions of 139H2 with R5 via residues E50/T59 in the heavy chain directly flank CDRH2 but are formally part of the framework regions, and both positions are mutated compared with the inferred germline precursor (see Fig 2). Two additional framework mutations in the heavy chain, that is, Y35 and T97, appear indirectly involved in MUC1 binding by positioning CDRH3 through hydrogen bonds with N106 and the backbone of Y111, respectively (see Fig S5A–C). Finally, the T4 residue of the APDTRPAPG epitope is a known glycosylation site, although 139H2

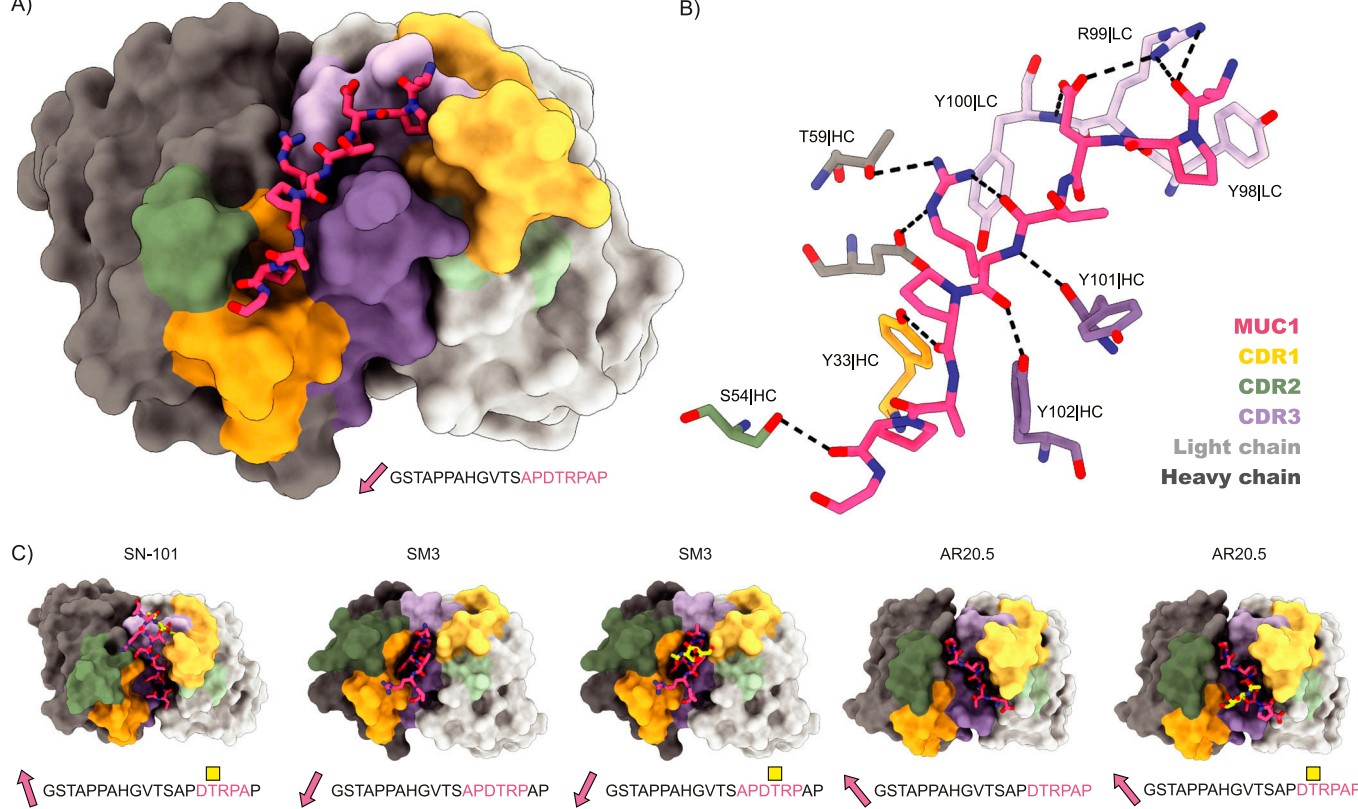

**Figure 4. Structure of 139H2 Fab in complex with the MUC1 peptide.**
**(A)** Surface representation of the Fab with complementarity-determining regions highlighted in colors and MUC1 peptide shown as a model. N- to C-terminus direction of MUC1 peptides is shown as a pink arrow. **(B)** Interactions between 139H2 Fab and MUC1 peptide. **(C)** Comparison with previously reported structures of monoclonal anti-MUC1 antibodies targeting the variable number of tandem repeats. Glycosylated residues of the epitope are depicted by a yellow square above.

binding is reported to be unaffected by the presence of a single *O*-linked GalNAc at this position (14, 25). The crystal structure reported here shows the T4 side chain to be pointing outward from the 139H2 paratope with no indication of potential clashes that would preclude binding of the epitope with glycosylated APDTRPAPG at the T4 position. In line with this previous report and our own structural data, we also found that 139H2 binds equally well to MUC1 reporter constructs with different types of *O*-linked glycans (see Fig 5A–F). The therapeutic potential of 139H2 therefore lies in targeting the characteristic overexpression of MUC1 in tumor cells, not the aberrant MUC1 glycosylation that is commonly described in MUC1-overexpressing cancers.

Comparison with previously reported structures of monoclonal anti-MUC1 antibodies targeting the VNTR reveals a striking diversity in the modes of binding (Fig 4C; a full overview of reported structures is listed in Table S3) (26 Preprint, 27, 28, 29, 30, 31, 32, 33, 34, 35, 36). Monoclonal antibodies 14A, 16A, and 5E5 all target a different region within the VNTR. Although monoclonal antibodies SM3, SN101, and AR20.5 all bind to the same immunodominant epitope of the VNTR as 139H2, the peptide is either shifted or oriented in the opposite direction relative to the cleft between the heavy and light chains. For SN101 and AR20.5, the peptide runs across this cleft in the opposite direction compared with 139H2. In SM3, the peptide is oriented in a similar direction but shifted by ~2 residues such that both D3 and R5 are contacting different CDRs. In contrast to 139H2,

each of the monoclonals compared above binds stronger to the glycosylated epitope. In the case of AR20.5 and SN101, this specificity can be explained by direct contacts made between the glycan and CDRs of the antibody. However, for SM3 the orientation of the glycosylated T4 residue is more similar to 139H2. In SM3, the GalNAc residue makes an additional hydrogen bond with a tyrosine in CDRL1. A similar interaction is predicted for 139H2, albeit through a different group of the GalNAc residue (see Fig S6).

## Discussion

Our study demonstrates how direct mass spectrometry–based protein sequencing enables the reconstruction of antibodies from hybridoma supernatants. In addition to recovering such precious resources for research and therapeutic applications, it also contributes to open and reproducible science by making the sequences of crucial monoclonal antibody reagents more readily available and accessible. Poorly defined (monoclonal) antibody products have notoriously been a challenge to reproducibility in life science research, and the present work shows that MS-based sequencing can offer helpful improvements in this regard (37, 38).

The reverse-engineered anti-MUC1 monoclonal antibody 139H2 reported here is suitable for Western blotting and immunofluorescence

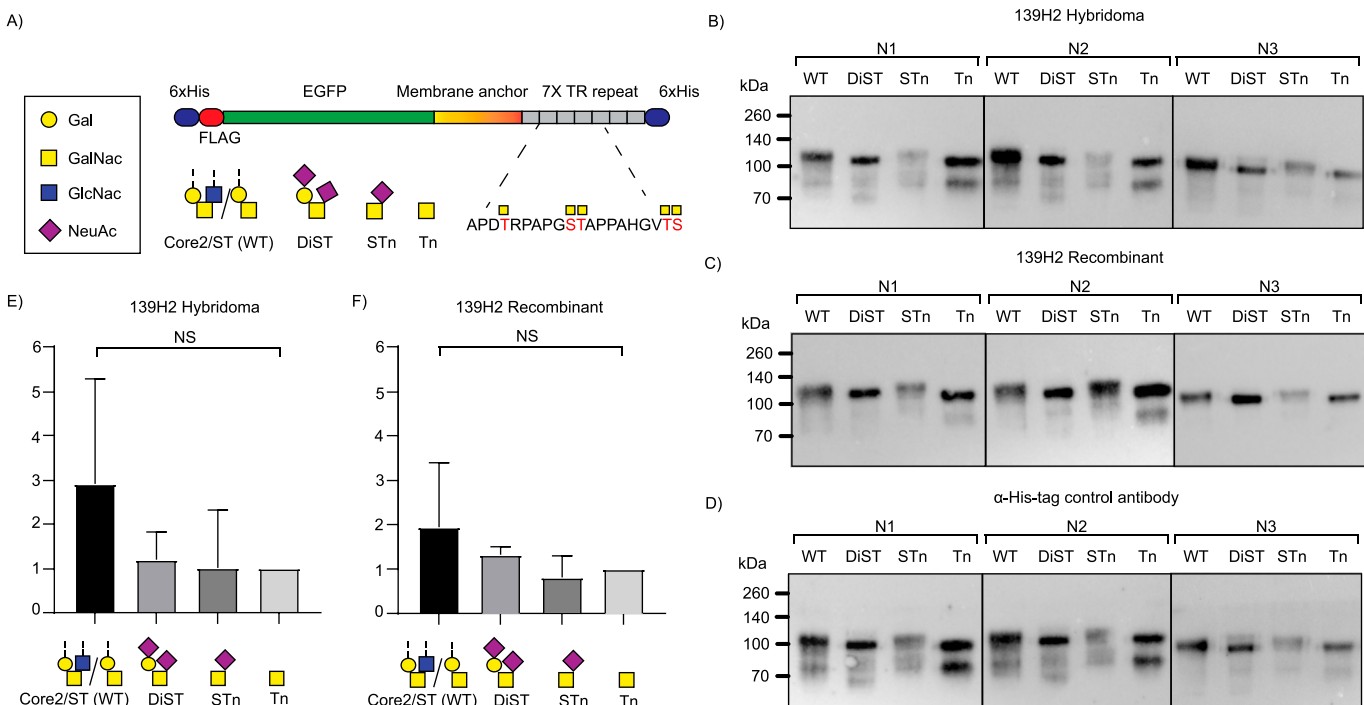

**Figure 5.   Binding of 139H2 to MUC1 reporter constructs with different O-linked glycosylation.**
**(A)** Schematic representation of the MUC1 fragments used. The four fragments used contain seven transmembrane repeats (TR) of MUC1 with 5 O-glycosylation sites with WT Core 2/ST (WT)/DiST/STn/Tn glycan structures. **(B)** Western blots against the MUC1 WT/DiST/STn/Tn fragments with the 139H2 hybridoma-derived antibody. **(C)** Western blots against the MUC1 WT/DiST/STn/Tn fragments with the 139H2 reverse-engineered antibody. **(D)** Western blots against the MUC1 WT/DiST/STn/Tn fragments with an α-His-tag antibody control. **(E, F)** Western blot band intensities analyzed with Image Lab 6.0 software. Calculated intensity ratios were made relative to the intensity of MUC1-Tn. No significant difference in binding of hybridoma-derived 139H2 or synthetic recombinant 139H2 was observed compared with the 6 α-His-tag antibody control.

microscopy and is likely suitable for other applications in FACS sorting of MUC1-positive cells, immunohistochemistry, and ELISA, as demonstrated for the original hybridoma-derived product ([16], [19], [20]). We show that 139H2 binds the immunodominant epitope of the VNTR in a unique way compared with previously described monoclonal antibodies against MUC1. Because of its previously reported glycan-independent binding, which we supported in this study by the determined structure in complex with the epitope, the 139H2 antibody is an important tool for current and future MUC1 research.

# Materials and Methods

### Purification of 139H2 from the hybridoma culture supernatant

The 139H2 in the hybridoma culture supernatant was a kind gift from John Hilkins from The Netherlands Cancer Institute (NKI). The 139H2 was purified with Protein G Sepharose 4 Fast Flow beads (Merck), washed with PBS, eluted with 0.2 mM glycine buffer, pH 2.5, neutralized with 1 M Tris–HCL, pH 8, and dialyzed against PBS with Pierce Protein Concentrators PES, 30 kD MWCO.

### Bottom-up proteomics—in-solution digestion

139H2 was denatured in 2% sodium deoxycholate, 200 mM Tris–HCl, and 10 mM Tris(2-carboxyethyl)phosphine, pH 8.0, at 95°C for 10 min, followed by 30-min incubation at 37°C for reduction. The samples were then alkylated by adding iodoacetic acid to a final concentration of 40 mM and incubated in the dark at RT for 45 min. 3 $\mu$g sample was then digested by trypsin (Promega) and elastase (Sigma-Aldrich) in a 1:50 ratio (w/w) in a total volume of 100 $\mu$l of 50 mM ammonium bicarbonate at 37°C for 4 h. After digestion, sodium deoxycholate was removed by adding 2 $\mu$l of formic acid (FA) and centrifuged at 14,000$g$ for 20 min. After centrifugation, the supernatant containing the peptides was collected for desalting on a 30-$\mu$m Oasis HLB 96-well plate (Waters). The Oasis HLB sorbent was activated with 100% acetonitrile and subsequently equilibrated with 10% formic acid in water. Next, peptides were bound to the sorbent, washed twice with 10% formic acid in water, and eluted with 100 $\mu$l of 50% acetonitrile/5% formic acid in water (vol/vol).

### Bottom-up proteomics—in-gel digestion

The hybridoma 139H2 was loaded on a 4–12% Bis-Tris precast gel (Bio-Rad) in non-reducing conditions and run at 120 V in 3-MOPS buffer (Bio-Rad). Bands were visualized with Imperial Protein Stain (Thermo Fisher Scientific), and the size of the fragments was evaluated by running a protein standard ladder (Bio-Rad). The IgG bands were cut and reduced by 10 mM Tris(2-carboxyethyl) phosphine at 37°C, followed by alkylation in 40 mM IAA at RT in the dark. The gel bands were digested by chymotrypsin and thermolysin at 37°C overnight in 50 mM ammonium bicarbonate buffer. The peptides were extracted with two-step incubation at RT in 50% ACN and 0.01% TFA, and then 100% ACN, respectively.

### Bottom-up proteomics—LC-MS/MS

The peptides obtained by in-solution and in-gel digestion were vacuum-dried and reconstituted in 100 $\mu$l of 2% FA. The digested peptides were separated by online reversed-phase chromatography on an Agilent 1290 Ultra-high-performance LC (UHPLC) or Dionex UltiMate 3000 (Thermo Fisher Scientific) coupled to a Thermo Fisher Scientific Orbitrap Fusion mass spectrometer. Peptides were separated using a Poroshell 120 EC-C18 2.7-$\mu$m analytical column (ZORBAX Chromatographic Packing, Agilent) and a C18 PepMap 100 trap column (5 mm × 300, 5 $\mu$m; Thermo Fisher Scientific). Samples were eluted over a 90-min gradient from 0% to 35% acetonitrile at a flow rate of 0.3 $\mu$l/min. Peptides were analyzed with a resolution setting of 60,000 in MS1. MS1 scans were obtained with a standard automatic gain control target, a maximum injection time of 50 ms, and a scan range of 350–2,000. The precursors were selected with a 3 m/z window and fragmented by sHCD and EThcD. The sHCD fragmentation included steps of 25%, 35%, and 50% normalized collision energies (NCE). EThcD fragmentation was performed with calibrated charge-dependent electron-transfer dissociation parameters and 27% NCE supplemental activation. For both fragmentation types, MS2 scans were acquired at a 30,000 resolution, a $4 \times 10^{-5}$ automatic gain control target, a 250-ms maximum injection time, and a scan range of 120–3,500.

### Bottom-up proteomics—peptide sequencing from MS/MS spectra

MS/MS spectra were used to determine de novo peptide sequences using PEAKS Studio X (version 10.6) ([39], [40]). We used a tolerance of 20 ppm and 0.02 Da for MS1 and MS2, respectively. Carboxymethylation was set as fixed modification of cysteine and variable modification of peptide N-termini and lysine. Oxidation of methionine and tryptophan and pyroglutamic acid modification of N-terminal glutamic acid and glutamine were set as additional variable modifications. The CSV file containing all the de novo–sequenced peptide was exported for further analysis.

### Bottom-up proteomics—template-based assembly via Stitch

Stitch (nightly version 1.4.0+802a5ba) was used for the template-based assembly ([41]). The mouse antibody database from IMGT was used as a template ([42]). The cutoff score for the de novo–sequenced peptide was set as 90, and the cutoff score for the template matching was set as 10. All the peptides supporting the sequences were examined manually.

### Cloning and expression of recombinant 139H2 IgG and Fab

To recombinantly express full-length anti-MUC1 antibodies, the proteomics sequences of both the light and heavy chains were reverse-translated and codon-optimized for expression in human cells using the Thermo Fisher Scientific web tool (https://www.thermofisher.com/order/gene-design/index.html). For the linker and Fc region of the heavy chain, the standard mouse Ig γ-1 (IGHG1) amino acid sequence (UniProt P01868.1) was used. An N-terminal secretion signal peptide derived from the human IgG

light chain (MEAPAQLLFLLLLWLPDTTG) was added to the N-termini of both heavy and light chains. BamHI and NotI restriction sites were added to the 5′ and 3′ ends of the coding regions, respectively. Only for the light chain, a double stop codon was introduced at the 3′ site before the NotI restriction site. The coding regions were subcloned using BamHI and NotI restriction–ligation into a pRK5 expression vector with a C-terminal octahistidine tag between the NotI site and a double stop codon 3′ of the insert so that only the heavy chain has a C-terminal AAAHHHHHHHH sequence for nickel-affinity purification (the triple alanine resulting from the NotI site). After the sequence was validated by Sanger sequencing, the HC/LC were mixed in a 1:1 DNA ratio and expressed in HEK293 cells by the ImmunoPrecise Antibodies (Europe) B.V. company. After expression, the culture supernatant of the cells was harvested and purified using a prepacked HisTrap Excel column (Cytiva), following the standard protocols (see Fig S2).

To recombinantly express anti-MUC1 Fab, the coding regions of the HC variable region were subcloned using AgeI and NheI restriction–ligation into a pRK5 expression vector. The subcloned region contains the mouse Ig γ-1 (IGHG1) Fab constant region with a C-terminal octahistidine tag followed by a double stop codon 3′ of the insert so that only the heavy chain has a C-terminal AAAHHHHHHHH sequence for nickel-affinity purification (the triple alanine resulting from the NotI site). After the sequence was validated by Sanger sequencing, the HC/LC were mixed in a 1:1 ($m/m$) DNA ratio and expressed in HEK293 cells by the ImmunoPrecise Antibodies (Europe) B.V. company. After expression, the culture supernatant was loaded onto a 5-ml HisTrap Excel column (Cytiva) using a peristaltic pump. A column was reconnected to the ÄktaGo system (Cytiva) for column wash (50 mM Tris at pH = 8, 150 mM NaCl) and step elution (50 mM Tris at pH = 8, 150 mM NaCl, 300 mM imidazole). Fractions from the peak corresponding to the Fab were concentrated using Amicon Ultra-15 (Millipore) and further purified by size-exclusion chromatography using Superdex 200 Increase 10/300 GL (Cytiva) in buffer containing 50 mM Tris (pH = 8), 150 mM NaCl.

## Mammalian cell lines and culture conditions

The human gastrointestinal epithelial cell lines HT29-MTX (43) and HT29-MTX ΔMUC1 (5) were cultured in 25-cm$^2$ flasks in DMEM containing 10% FCS at 37°C in 10% CO$^2$.

## Western blot

HT29-MTX and HT29-MTX ΔMUC1 lysates were prepared from cells grown to full confluency for 7 d in a six-well plate. Cells were harvested by scraping and lysed with lysis buffer (10% SDS in PBS with 1× Halt Protease Inhibitor Cocktail). The concentration was measured by BCA assay, 5× Laemmli buffer was added, and the sample was boiled for 15 min at 95°C. A mucin–SDS gel was made according to reference 5; 40 $\mu$g of protein was added to each well and run in boric acid–Tris buffer (192 mM boric acid, Merck; 1 mM EDTA, Merck; and 0.1% SDS, to pH 7.6 with Tris) at 25 mA for 1.5 h. Proteins were transferred to a polyvinylidene fluoride (PVDF) membrane using wet transfer for 3 h at 90 V/4°C in transfer buffer

(25 mM Tris; 192 mM glycine, Merck; and 20% methanol, Merck). Afterward, membranes were blocked with 5% BSA in TSMT (20 mM Tris; 150 mM NaCl, Merck; 1 mM CaCl$_2$ [Sigma-Aldrich]; 2 mM MgCl$_2$, Merck; adjusted to pH 7 with HCl; and 0.1% Tween-20 [Sigma-Aldrich]) overnight at 4°C. The next day, membranes were washed with TSMT and incubated with 139H2 WT, synthetic, or FAB antibodies (1:1,000) in TSMT containing 1% BSA for 1 h at RT. Membranes were washed again with TSMT and incubated with α-mouse IgG secondary antibody (A2304; Sigma-Aldrich) diluted 1:8,000 in TSMT with 1% BSA for 1 h at RT, and washed with TSMT followed by TSM. For detection of actin, cell lysates were loaded onto a 10% SDS–PAGE, transferred to PVDF membranes, and incubated with α-actin antibody (1:2,000; bs-0061R; Bioss) and α-rabbit IgG (1: 10,000; A4914; Sigma-Aldrich). Blots were developed with the Clarity Western ECL kit (Bio-Rad) and imaged in a Gel-Doc system (Bio-Rad).

## Western blot of MUC1 reporter constructs

Four MUC1 reporter constructs, expressed in engineered HEK293 cells, were a kind gift from Christian Büll of the Copenhagen Center for Glycomics. Each reporter construct in 1×, PBS was boiled in 5× Laemmli buffer. 10 ng/25 ng of each construct was loaded per well on a 10% Bis–acrylamide SDS gel for the 139H2/6× His-tag blots, respectively. Samples were run in 1× Novex Tris–Glycine SDS Running Buffer (Thermo Fisher Scientific) for 1.5 h at 120 V. Proteins were transferred to a 0.2-$\mu$m Trans-Blot PVDF membrane (Bio-Rad) and transferred at 1.3 A/25 V for 7 min using the Trans-Blot Turbo system (Bio-Rad). Afterward, membranes were blocked with 5% BSA in TSMT (20 mM Tris; 150 mM NaCl, Merck; 1 mM CaCl$_2$, Sigma-Aldrich; 2 mM MgCl$_2$, Merck, adjusted to pH 7 with HCl; and 0.1% Tween-20, Sigma-Aldrich) overnight at 4°C. The next day, membranes were washed with TSMT and incubated with 139H2 WT, synthetic antibody (1:1,000) or HisProbe-HRP Conjugate (15165, 1:5,000; Thermo Fisher Scientific) in TSMT containing 1% BSA for 1 h at RT. The 6× His-tag blots were washed with TMST and TSM, developed with the Clarity Western ECL kit (Bio-Rad), and imaged in a Gel-Doc system (Bio-Rad). The 139H2 membranes were washed again with TSMT and incubated with α-mouse IgG secondary antibody (A2304; Sigma-Aldrich) diluted 1:8,000 in TSMT with 1% BSA for 1 h at RT, washed with TSMT followed by TSM, and developed.

## Confocal microscopy

HT29-MTX and HT29-MTX ΔMUC1 cells were grown for 7 d to reach a confluent monolayer on coverslips (8 mm diameter #1.5) in 24-well plates. Cells were washed with Dulbecco's Phosphate-Buffered Saline (DPBS, D8537) and fixed with 4% paraformaldehyde in PBS (Affymetrix) for 30 min at RT. Fixation was stopped by adding 50 mM NH$_4$Cl in PBS for 15 min. Cells were washed two times and permeabilized in binding buffer (0.1% saponin [Sigma-Aldrich] and 0.2% BSA [Sigma-Aldrich] in DPBS) for 30 min. Coverslips were incubated with 139H2 WT, synthetic FAB at 1:100 dilution for 1 h, washed 3× with binding buffer, and incubated with Alexa Fluor 488–conjugated α-mouse IgG secondary antibodies (1:200; A11029; Thermo Fisher Scientific) and DAPI at 2 $\mu$g/ml (D21490; Invitrogen) for 1 h. Coverslips were washed 3× with DPBS, desalted in Milli-Q,

dried and embedded in ProLong Diamond Mounting Solution (Thermo Fisher Scientific), and allowed to harden. Images were collected on a Leica SPE-II confocal microscope with a 63× objective (NA 1.3, HCX PLANAPO oil) and controlled by Leica LAS AF software with default settings to detect DAPI, Alexa Fluor 488, Alexa Fluor 568, and Alexa Fluor 647. Axial series were collected with step sizes of 0.29 $\mu$m.

### SPR

N-terminally biotinylated synthetic MUC1 peptide with the sequence biotin-GGS-APDTRPAPG was ordered from GenScript. This was dissolved in PBS and printed on a planar streptavidin-coated SPR chip (P-Strep, SSens B.V.) using a continuous flow microfluidics spotter (Wasatch), flowing for 1 h at RT, after which it was washed with SPR buffer (150 mM NaCl, and 25 mM 4-(2-hydroxyethyl)-1-piperazineethanesulfonic acid [Hepes] with 0.005% Tween-20) for 15 min and quenched with biotin solution (10 mM biotin in SPR buffer). SPR experiments were performed using an IBIS-MX96 system (IBIS Technologies) with SPR buffer as the running buffer. A dilution series of 2× steps of the full recombinant 139H2 or Fab were prepared, starting from a 10.0-$\mu$M stock for full IgG and a 7.88-$\mu$M stock for the Fab, diluting with SPR buffer. 20 dilution steps (including the stock) were used for the full IgG, and 10 dilutions were used for the Fab. SPR experiments were performed as a kinetic titration without regenerating in between association/dissociation cycles, with 30-min association and 10-min dissociation time for the full IgG and 6-min association and 4-min dissociation for the Fab. Binding affinity was determined by fitting data at binding equilibrium to a two-site binding model for the full IgG and a one-site (Langmuir) binding model for the Fab, using Scrubber 2.0 (BioLogic Software) and GraphPad Prism 5 (GraphPad Software, Inc.).

### Crystallization and data collection

Sitting-drop vapor diffusion crystallization trials were set up at 20°C by mixing 150 nl of complex with 150 nl of reservoir solution. The complex sample consisted of purified 139H2 Fab and MUC1 epitope peptide (APDTRPAPG; GenScript) in a 1:2.5 M ratio, at a total concentration of 3.8 mg/ml in a buffer of 50 mM trisaminomethane at pH 8.0 and 150 mM NaCl. The diffracting crystals grew in a condition of 0.2 M NaCl, 0.1 M sodium–phosphocitrate, and 20% wt/vol polyethylene glycol (PEG) 8,000 used as reservoir solution. A 3:1 mixture of reservoir solution and glycerol was added as a cryoprotectant to the crystals before plunge-freezing them in liquid nitrogen. Datasets were collected at 100 K at Diamond Light Source beamline I24, equipped with an Eiger 9M detector (Dectris), at a wavelength of 0.6199 Å.

### Structure determination and refinement

Collected datasets were integrated using the xia2.multiplex pipeline (44), and the three best datasets were subsequently merged and scaled in AIMLESS to a maximum resolution of 2.5 Å. The resolution limit cutoff was determined based on the mean intensity correlation coefficient of half-datasets, $CC_{1/2}$. An initial model of 139H2 Fab was generated using ColabFold (45). The variable region and constant region were placed in subsequent PHASER (46) runs, the short linkers between the two regions were built manually, and the CDRs were adjusted in COOT (47). Clear density for the MUC1 peptide was present in the Fo-Fc map, and the peptide was built manually in COOT. The structure was refined by iterative rounds of manual model building in COOT and refinement in REFMAC5 (48). The final model was assessed using MolProbity (49). All programs were used as implemented in CCP4i2 version 1.1.0 (50).

## Data Availability

The raw LC-MS/MS files and analyses have been deposited to the ProteomeXchange Consortium via the PRIDE partner repository with the dataset identifier PXD043489. Coordinates and structure factors for 139H2 bound to the MUC1 epitope peptide have been deposited to the Protein Data Bank with the accession code 8P6I. The plasmids for 139H2 expression in mammalian cells are made available through Addgene, under plasmid ID 206201 and 206202 for the heavy and light chains, respectively.

## Supplementary Information

## Acknowledgements

This research was funded by the Dutch Research Council NWO Gravitation 2013 BOO, Institute for Chemical Immunology (ICI; 024.002.009), to J Snijder. K Strijbis and KCAP Giesbers are supported by the European Research Council under the European Union's Horizon 2020 research and innovation program (ERC-2019-STG-852452). The authors would like to thank Diamond Light Source for beamtime (proposal mx25413) and the staff of beamline I24 for assistance with data collection.

### Author Contributions

W Peng: formal analysis, investigation, visualization, methodology, and writing—original draft, review, and editing.
KCAP Giesbers: formal analysis, investigation, visualization, methodology, and writing—review and editing.
M Šiborová: formal analysis, investigation, visualization, methodology, and writing—review and editing.
JW Beugelink: formal analysis, investigation, visualization, methodology, and writing—review and editing.
MF Pronker: formal analysis, investigation, visualization, methodology, and writing—review and editing.
D Schulte: data curation, software, and writing—review and editing.
J Hilkens: resources.
BJC Janssen: resources, supervision, investigation, and writing—review and editing.
K Strijbis: conceptualization, resources, supervision, funding acquisition, investigation, and writing—review and editing.

J Snijder: conceptualization, resources, supervision, funding acquisition, investigation, and writing—original draft, review, and editing.

## Conflict of Interest Statement

The authors declare that they have no conflict of interest.

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
