## [Reviewer comments · Life Science Alliance]

Life Science Alliance

Reverse engineering the anti-MUC1 antibody 139H2 by mass spectrometry-based de novo sequencing

Weiwei Peng, Koen Giesbers, Marta Siborova, J. Beugelink, Matti Pronker, Douwe Schulte, John Hilkens, Bert Janssen, Karin Strijbis, and Joost Snijder

DOI: <https://doi.org/10.26508/lsa.202302366>

Corresponding author(s): Joost Snijder, Utrecht University and Karin Strijbis, Utrecht University

Review Timeline:

Submission Date:	2023-09-12
Editorial Decision:	2023-12-27
Revision Received:	2024-03-06
Editorial Decision:	2024-03-07
Revision Received:	2024-03-11
Accepted:	2024-03-12

Transaction Report:

December 27, 2023

Re: Life Science Alliance manuscript #LSA-2023-02366

Joost Snijder
Utrecht University
Padualaan 8
Utrecht 3584CH
Netherlands

Dear Dr. Snijder,

Thank you for submitting your manuscript entitled "Reverse engineering the anti-MUC1 hybridoma antibody 139H2 by mass spectrometry-based de novo sequencing" to Life Science Alliance. The manuscript was assessed by expert reviewers, whose comments are appended to this letter. We invite you to submit a revised manuscript addressing the Reviewer comments.

Thank you for this interesting contribution to Life Science Alliance. We are looking forward to receiving your revised manuscript.

Sincerely,

B. MANUSCRIPT ORGANIZATION AND FORMATTING:

Reviewer #1 (Comments to the Authors (Required)):

Peng et al. identified the sequence of hybridoma-derived 139H2 antibody through mass spectrometry-based sequencing. They also demonstrated that the recombinant 132H2 antibody retains the binding capacity and can be used for western blot and immunofluorescence microscope as comparable as hybridoma-derived 139H2 antibody. Binding was characterized using SPR and structural analysis was performed with recombinant 132H2 Fab. This is a comprehensive and well written manuscript. The result is systematically discussed, and it strongly supports the conclusion. I only have a couple of minor comments.

1. Authors described the interaction of 132H2 Fab and MUC1 peptide and compared it with the interaction of MUC1 peptide with different anti-MUC1 antibodies. Is there a clinical significance of 132H2 antibody based on its unique interaction with MUC1 peptide?
2. In Figure 4 legend, (B) 'Interactions of' needs to be deleted.

Reviewer #2 (Comments to the Authors (Required)):

The authors demonstrate the reverse engineering and validation of an anti-MUC1 antibody. The authors describe de novo sequencing from hybridoma, construct generation, validation by WB and SPR, structure elucidation and then eiptope mapping. The paper is clearly written and describes experiments in detail. My only issue with the manuscript is it's lack of novelty. The deposition of the plasmid for public use is indeed beneficial to the scientific community, and the manuscript serves as a classic roadmap for de novo antibody generation and validation and as such is valuable, but each step of the process has been described previously and is common practice in industrial settings, so I was left feeling slightly disappointed after reading.

Why choose the 'APD' peptide from the VNTR as your target for Ab de novo reverse engineering (as opposed to another region)? What was the importance of targeting this common epitope? I'm surprised that glycosylation in the eiptope region does not affect binding, but the authors supply rationale for this based on the direction of T4.

Would the authors consider performing functional/activity based assays as validation in addition to the binding assays performed? Assessment of activity in addition to binding would further confirm the correctness of the recombinant Ab exercise.

Minor edits:

Abstract Line 27 need period after adenocarcinomas

Abstract line 28 has 2 periods at sentence end

bottom up proteomics line 262 describes alkylolation twice

We would like to thank the reviewers for their kind and helpful comments. We were happy to read that they characterized the work as 'comprehensive and well written', that the results 'strongly support the conclusions', and that it serves as 'a classic roadmap for de novo antibody generation and validation'. We provided a point-by-point reply to the reviewers' comments below.

Reviewer #1 (Comments to the Authors (Required)):

Peng et al. identified the sequence of hybridoma-derived 139H2 antibody through mass spectrometry-based sequencing. They also demonstrated that the recombinant 132H2 antibody retains the binding capacity and can be used for western blot and immunofluorescence microscope as comparable as hybridoma-derived 139H2 antibody. Binding was characterized using SPR and structural analysis was performed with recombinant 132H2 Fab. This is a comprehensive and well written manuscript. The result is systematically discussed, and it strongly supports the conclusion. I only have a couple of minor comments.

We are happy to read that the reviewer appreciates the work and only has minor comments.

1. Authors described the interaction of 132H2 Fab and MUC1 peptide and compared it with the interaction of MUC1 peptide with different anti-MUC1 antibodies. Is there a clinical significance of 132H2 antibody based on its unique interaction with MUC1 peptide?

The 139H2 hybridoma was raised against breast cancer cell derived plasma membranes, with diagnostic and therapeutic applications in mind. In the introduction of the revised manuscript, we write:

"Since the late 1980's, several monoclonal antibodies against MUC1 have been described and explored for the diagnosis and treatment of MUC1 overexpressing cancers^{15,16}. Peptide mapping experiments have revealed that many such monoclonal antibodies target a similar region within the VNTR of MUC1, resulting in the definition of

an immunodominant peptide corresponding to the subsequence APDTRPAP¹⁷. One such antibody is 139H2, a hybridoma monoclonal antibody that was raised against human breast cancer plasma membranes^{15,16}. In different studies, 139H2 has been applied for the diagnostics of MUC1-overexpressing cancers and radioimmunotherapy^{15,16,18}.”

However, as demonstrated before and confirmed in our study, 139H2 stands out compared to other MUC1-directed antibodies for its glycosylation-*independent* binding to the MUC1 VNTR. While tumor-associated MUC1 distinguishes itself from MUC1 in healthy tissue by the presentation of shorter O-glycans, 139H2 binding is unaffected by these differences. The therapeutic potential of 139H2 is therefore mediated only by the characteristic overexpression of MUC1 in tumor cells, not its aberrant glycosylation. This is now further clarified in the corresponding result section, where we write:

“The therapeutic potential of 139H2 therefore lies in targeting the characteristic overexpression of MUC1 in tumor cells, but not the aberrant MUC1 glycosylation that is commonly described in MUC1-overexpressing cancers.”

2. In Figure 4 legend, (B) 'Interactions of' needs to be deleted.

Done

Reviewer #2 (Comments to the Authors (Required)):

The authors demonstrate the reverse engineering and validation of an anti-MUC1 antibody. The authors describe de novo sequencing from hybridoma, construct generation, validation by WB and SPR, structure elucidation and then epitope mapping. The paper is clearly written and describes experiments in detail. My only issue with the manuscript is its lack of novelty. The deposition of the plasmid for public use is indeed beneficial to the scientific community, and the manuscript serves as a classic roadmap for de novo antibody generation and validation and as such is valuable, but each step of the process has been described previously and is common practice in industrial settings, so I was left feeling slightly disappointed after reading.

We were happy to read that the reviewer appreciates our work as ‘a classic roadmap for de novo antibody generation and validation’. We kindly disagree with the reviewer’s characterization that the work lacks novelty because this would be common practice in industrial settings. While it may be the case that MS-based antibody sequencing is more widely applied in industry, the public scientific record of this is rather sparse, with only a handful of papers describing the full process and quite a few of those sparse papers published by us. Our current paper describes the state-of-the-art in proteomics-based antibody sequencing, using as little as 12 micrograms of input material to obtain a functional reverse-engineered antibody, using publicly available tools. While we consider the work to be a powerful showcase of the technology, we believe the true value and impact of the paper is that we made 139H2 a publicly available resource for MUC1 research. As it stands, it is now the only publicly available antibody with glycosylation-independent MUC1 binding, with a fully defined sequence and an atomically defined epitope.

Why choose the 'APD' peptide from the VNTR as your target for Ab de novo reverse engineering (as opposed to another region)? What was the importance of targeting this common epitope? I'm surprised that glycosylation in the epitope region does not affect binding, but the authors supply rationale for this based on the direction of T4.

The 139H2 antibody was raised against breast cancer derived plasma membranes. Targeting the VNTR was not a design choice in raising the hybridoma, but rather a consequence of the immunodominance of this epitope (likely because of the many repeats with which it is presented in a full MUC1 ectodomain). As we write in the introduction, peptide mapping experiments have previously pinpointed this immunodominant epitope as the target of 139H2 (as well as many other MUC1-directed antibodies):

“Since the late 1980’s, several monoclonal antibodies against MUC1 have been described and explored for the diagnosis and treatment of MUC1 overexpressing cancers^{15,16}. Peptide mapping experiments have revealed that many such monoclonal antibodies target a similar region within the VNTR of MUC1, resulting in the definition of an immunodominant peptide corresponding to the subsequence APDTRPAP¹⁷. One

such antibody is 139H2, a hybridoma monoclonal antibody that was raised against human breast cancer plasma membranes^{15,16}.”

The glycosylation-independent binding of 139H2 to MUC1 is indeed a striking feature. This has been previously described in refs 15/16 and is also confirmed by us in the current work based on Western blots of the MUC1 reporter constructs. Luckily, we indeed managed to provide a convincing rationale for this binding property, as our crystal structure revealed the T4 sidechain to be pointing away from the 139H2 paratope region.

Would the authors consider performing functional/activity based assays as validation in addition to the binding assays performed? Assessment of activity in addition to binding would further confirm the correctness of the recombinant Ab exercise.

Previous work with 139H2 has never described any functional/activity effects on MUC1. It has rather been used as a targeting probe for diagnostic/therapeutic purposes, and especially as an affinity reagent in MUC1 research. It was therefore not trivial to come up with a suitable functional/activity-based assay as requested by the reviewer. After reviewing the literature, we encountered a study by Zhao et al. 2010 in *Molecular Cancer* (<https://doi.org/10.1186/1476-4598-9-154>) that described induced aggregation of cancer cells by a MUC1-specific antibody. We tried to reproduce this effect with 139H2 (as measured by light microscopy), comparing aggregation of breast cancer cells with/without induction of MUC1 expression, in the presence/absence of 139H2 and a negative control antibody that targets the cytoplasmic tail of MUC1, but could not observe any induced cell aggregation by either the hybridoma derived or reverse-engineered 139H2:

Given the negative result, we chose not to include this in our revised manuscript. We would like to reiterate that the main use of 139H2 is as an affinity reagent for MUC1 research, with potential therapeutic/diagnostic applications. As such, the reverse-engineered 139H2 is thoroughly validated and a useful resource for the MUC1 research community.

Minor edits:

Abstract Line 27 need period after adenocarcinomas

Done

Abstract line 28 has 2 periods at sentence end

Done

bottom up proteomics line 262 describes alkylation twice

Done

March 7, 2024

RE: Life Science Alliance Manuscript #LSA-2023-02366R

Dr. Joost Snijder
Utrecht University
Padualaan 8
Utrecht 3584CH
Netherlands

Dear Dr. Snijder,

Thank you for submitting your revised manuscript entitled "Reverse engineering the anti-MUC1 antibody 139H2 by mass spectrometry-based de novo sequencing". We would be happy to publish your paper in Life Science Alliance pending final revisions necessary to meet our formatting guidelines.

- please be sure that the authorship listing and order is correct
- please add ORCID ID for the secondary corresponding author -- they should have received instructions on how to do so
- please use the [10 author names et al.] format in your references (i.e., limit the author names to the first 10)
- please add callouts for Figures 3A-B; 4A-C; 5A-F; S2A-B; S3A-B; S4A-B; S5A-C to your main manuscript text;

A. FINAL FILES:

B. MANUSCRIPT ORGANIZATION AND FORMATTING:

Sincerely,

March 12, 2024

RE: Life Science Alliance Manuscript #LSA-2023-02366RR

Dr. Joost Snijder
Utrecht University
Padualaan 8
Utrecht 3584CH
Netherlands

Dear Dr. Snijder,

Thank you for submitting your Resource entitled "Reverse engineering the anti-MUC1 antibody 139H2 by mass spectrometry-based de novo sequencing". It is a pleasure to let you know that your manuscript is now accepted for publication in Life Science Alliance. Congratulations on this interesting work.

DISTRIBUTION OF MATERIALS:

Again, congratulations on a very nice paper. I hope you found the review process to be constructive and are pleased with how the manuscript was handled editorially. We look forward to future exciting submissions from your lab.

Sincerely,
